# Correlation between Dietary Intake of Vitamins and Oral Health Behaviors: A Cross-Sectional Study

**DOI:** 10.3390/ijerph20075243

**Published:** 2023-03-23

**Authors:** Moeka Ariizumi, Maya Izumi, Sumio Akifusa

**Affiliations:** 1School of Oral Health Sciences, Faculty of Dentistry, Kyushu Dental University, Fukuoka 803-8580, Japan; 2Division of Health Promotion, Department of Public Health and Medical Care, Hyogo Prefectural Government, Kobe 650-8567, Japan

**Keywords:** oral health behavior, vitamins, national health and nutrition examination survey, national survey of dental diseases

## Abstract

This study aimed to investigate whether oral health behaviors were related to the dietary intake of vitamins. In this cross-sectional study, we included respondents of the 2016 national health and nutrition examination survey, and dental diseases from Hyogo Prefecture, Japan. Data on sociodemographic characteristics, findings of blood tests related to metabolic syndrome, dietary intake, oral health status, and behaviors were collected. Participants were divided into two groups based on their oral health behavior: the yes group (performed interdental cleaning or tongue brushing) and the no group (did not perform the behaviors). The study included 218 participants (male: 107, female: 111) aged 64.5 (range, 22–93) years. There were 133 (61.0%) and 85 (39.0%) participants in the yes and no groups, respectively. The daily intake of vitamins A, B_2_, B_6_, E, and K, folic acid, and niacin in the yes group was significantly higher than that in the no group. Oral health behavior correlated with the intake of vitamin B_2_ (*p* = 0.029), folic acid (*p* = 0.006), and vitamin K (*p* = 0.043) after adjusting for possible confounders. Oral health behavior (interdental cleaning or tongue brushing) correlated with the daily intake of vitamins B_2_, K, and folic acid.

## 1. Introduction

Vitamin deficiencies cause serious diseases, such as nyctalopia, beriberi, scurvy, pernicious anemia, pellagra, and rachitis, and are almost avoided in developed countries in persons without underlying diseases, such as cancers [1,2]. Recently, it was noted that in developed countries, moderate insufficiency of vitamins compared to deficiency increases the risk of various diseases [3,4,5]. The definition of malnutrition by the World Health Organization (WHO) includes micronutrient (vitamins and minerals) deficiencies or insufficiencies. The results of a national health and nutrition examination survey in Japan demonstrated that compared with the recommended dietary allowance (RDA) or adequate intake, the dietary intake of vitamin D, niacin, folic acid, and pantothenic acid was 100% sufficient, but that of vitamins A, C, E, and B vitamins was inadequate in adults aged 30–39 years [6]. Insufficiency of vitamin D (calciferol) is related to an increased risk of non-communicable diseases, such as obesity [7], cardiovascular diseases [8], cancer [9], and multiple sclerosis [10]. Insufficiency of vitamin B_12_ (cyanocobalamin) is related to the risk of cardiovascular diseases; moreover, folate is linked to fracture risk, and vitamin B_6_ (pyridoxine) is linked to cognitive decline [11]. Insufficiencies of vitamin B_6_ and B_12_ are linked to an elevation in plasma homocysteine levels, which is associated with the risk of cardiovascular diseases. In older adults, insufficiency of vitamin B_1_ (thiamine) increases the risk of heart failure. In acute respiratory tract infections, including coronavirus disease 2019 (COVID-19) and influenza, supplementation with vitamins C and D potentially reduces the risk of infection.

The relationship between oral health behavior and eating habits is well known, especially in dental caries [12,13]. Although eating habits are not determining indicators for dental caries because of multifactorial disease, vegetable or greens consumption is inversely correlated with caries development [14,15,16,17]. A recent study demonstrated an association between higher adherence to the dietary approaches to stop hypertension or Mediterranean diets and a lower risk of periodontal diseases [18]. Regarding the relationship between a healthy lifestyle and diet choices, a previous study demonstrated that the healthy lifestyle cluster (no smoking, limited alcohol consumption, nutrition, exercise, and weight control) was associated with fruit and vegetable consumption [19] and was closely linked to vitamin intake. Nevertheless, to the best of our knowledge, there are no studies on the relationship between food choice linked to dietary intake of vitamins and oral health behavior. We hypothesized that good oral health-related behavior is related to good eating habits (i.e., selecting food enriched with micronutrients including vitamins). Thus, this study aimed to investigate whether oral health behavior is associated with the daily intake of vitamins using data from a national survey in Japan.

## 2. Materials and Methods

### 2.1. Data Collection

The participants in this cross-sectional study were the respondents of the national health and nutrition examination survey and the national survey of dental diseases in 2016 in Hyogo Prefecture, Japan. For statistical analysis, we obtained permission for the secondary use of data from the two national surveys from the Ministry of Health, Labour and Welfare under the authorization of the governor of the prefecture. The respondents for this study were ≥20 years old. The respondents for the national health and nutrition examination survey were selected by a family unit. Families with a foreign householder, or those who relied on mass feeding for all their daily meals, were excluded. The respondents who consumed liquid meals or who were always separated from their family were also excluded. Data from the two surveys were matched using the respondents’ ID. Information on the educational level of the area of residence of the respondents was collected from the national census. The education level was divided based on the percentage of those entering higher education (university, junior college, and technical college). Data on the income level of the area of residence of the respondents were collected from the basic survey on wage structure. Because the data were obtained for secondary use and used upon obtaining governmental approval, the study did not require ethical review.

### 2.2. National Health and Nutrition Examination Survey in 2016

The data included the respondents’ ID, age, sex, height, weight, abdominal circumference, blood pressure (systolic arterial and diastolic blood pressure), blood test according to the procedure (ratio of HbA1c, concentration of total cholesterol, high-density lipoprotein [HDL] cholesterol, and low-density lipoprotein cholesterol), smoking habit, and dietary intake (total protein, total fat, carbohydrate, sodium, calcium, magnesium, vitamins A, B_1_, B_2_, B_6_, B_12_, C, D, E, and K, folic acid, and niacin). The survey of dietary intake was carried out by weighing and recording the food consumed by the respondents for each family unit.

### 2.3. National Survey of Dental Diseases in 2016

The data in the survey included the respondent ID, age, sex, worries about the condition of the mouth and teeth, daily frequency of tooth brushing, methods for oral care other than toothbrushing, number of teeth, periodontal pocket depth evaluation using the WHO periodontal probe, and bleeding on probing (BOP). Probing was performed using the WHO probe by sliding the probe tip along the surface of the teeth with light pressure (20 g). BOP was considered to be present if observed within 10–30 s after probing.

### 2.4. Sample Size

This study included 218 participants, from the national survey for Hyogo prefecture.

### 2.5. Outcomes

The primary outcome was the daily intake of vitamins based on oral health behavior, excluding toothbrushing. The secondary outcome was the correlation between the daily intake of each vitamin and oral care behavior.

### 2.6. Statistical Analysis

Values are shown as the median (minimum–maximum) for continuous variables and number (%) for categorized variables. For statistical analysis, the participants were divided into two groups based on their oral care behavior other than toothbrushing, i.e., the yes group used interdental cleaning tools to care for their mouth and regular tongue cleaning, among other methods. The no group consisted of participants who did not use interdental cleaning tools as the yes group did. We used the Mann–Whitney U test for continuous variables and the chi-squared for categorized variables. When a category included 3 or more items, residual analysis was performed. To examine the correlation between the daily intake of each nutrient and oral care status, multiple regression analysis was performed. To analyze the collinearity of the explanatory variables for multiple regression analysis, we assessed the variance inflation factor (VIF). We performed all analyses, expect for power analysis, using SPSS version 28 (IBM, Inc., Tokyo, Japan). Power analysis was performed using G*power [20]. All statistical comparisons were two-sided, and <5% error was considered statistically significant. In residual analysis, when the absolute value of the adjusted standard residual was over 1.96, the p-value was judged as being <5%.

## 3. Results

Overall, 218 participants were included in this study (male: 107, female: 111), and the mean age was 64.5 (22–93) years. There were 133 (61.0%) and 85 (39.0%) participants in the yes and no groups, respectively. Table 1 shows the participants’ characteristics divided by group. There were no significant differences in age, sex, and smoking habits between the groups. In terms of the education level of the settled area, the percentage of those who entered higher education of the no group (44.7%) was significantly higher than that of the yes group (25.6%, residual: 2.9, chi-squared test: *p* = 0.007). Similarly, in terms of the income level of the settled area, the percentage of the first grade (lower income) of the no group (37.6%) was significantly higher than that of the yes group (18.8%, residual: 3.1, and chi-squared test: *p* = 0.004). Regarding oral health status, only the number of teeth showed a significant difference. In terms of the physical examination, there were no differences between the groups. The plasma HDL cholesterol concentration of the yes group (65.0 [29–112] mg/dL) was significantly higher than that of the no group (60.0 [33–99] mg/dL, *p* = 0.028). Results for the comparison of daily nutrient intake (yes vs. no group, respectively) was as follows: total protein (470.7 [43.7–7713.5] g vs. 355.4 [41.0–2916.2] g, *p* = 0.001), total fat (62.2 [20.3–141.3] g vs. 56.2 [15.9–121.4] g, *p* = 0.034), calcium (580.7 [113.0–1282.1] mg vs. 491.6 [116.7–1922.8] mg, *p* = 0.016), vitamin A (470.7 [43.7–7713.5] µgRAE vs. 355.4 [41.0–2916.2] µgRAE, *p* = 0.001), vitamin B_2_ (1.2 [0.3–2.9] mg vs. 1.1 [0.2–2.2] mg, *p* = 0.028), vitamin B_6_ (1.2 [0.3–3.0] vs.1.0 [0.3–2.8] mg, *p* = 0.017), vitamin E (7.1 [2.0–17.8] mg vs. 6.5 [1.5–15.3] mg, *p* = 0.016), vitamin K (197.8 [31.8–926.5] µg vs. 145.4 [16.4–743.8] µg, *p* = 0.005), folic acid (294.5 [87.1–1262.6] µg vs. 252.5 [102.4–734.3] µg, *p* = 0.003), and niacin (14.3 [3.0–45.5] mgNE vs. 12.5 [3.7–41.9] mgNE, *p* = 0.008). A significant correlation between each vitamin intake was observed among all vitamins that showed significant differences between the evaluated groups (Appendix A).

Next, to examine the correlation between the intake of nutrients that indicated statistical differences between the groups and oral care behavior, we performed multiple regression analysis (Table 2). Because the VIFs of the education and income levels of the settled area were high (above 3.0), we deleted income level from the multiple regression analysis. Oral care behavior was correlated with daily intake of vitamin B_2_ (β = 0.16, *p* = 0.029), folic acid (β = 0.20, *p* = 0.006), and vitamin K (β = 0.15, *p* = 0.043) after adjusting for age, sex, education level of the settled area, number of teeth, and abdominal circumference. For the multiple regression analysis, when the effect size (f 2) and β/α ratio were 0.15 and 4, respectively, power was calculated as 0.989.

## 4. Discussion

We found that oral care behavior (e.g., interdental cleaning or tongue brushing, excluding toothbrushing) individually correlated with the intake of vitamins B_2_, K, and folic acid. Individuals who do not engage in these oral care behaviors may have a lower intake of those vitamins. Vitamin B_2_ (riboflavin) is related to energy metabolism, such as the tricarboxylic acid cycle, electron transport, and β–oxidation. Vitamin B_2_ deficiency has an impact on iron absorption, the metabolism of tryptophan, mitochondrial dysfunction, the gastrointestinal tract, and brain dysfunction [21]. Regarding oral-related disorders, vitamin B_2_ deficiency is associated with stomatitis, perleche, glossitis, and seborrheic dermatitis, resolving with correction of hypo nutrition of vitamin B_2_ [22]. A recent study demonstrated that a lower serum level of vitamin B_2_ increased the risk of radiation-induced mucositis among nasopharyngeal carcinoma patients who underwent radiotherapy [23]. Vitamin B_2_ has several biological benefits, including antioxidant [24], anti-aging [25], anti-inflammatory [26], antinociceptive [27], and anticancer properties [28,29]. Low intake of vitamin B_2_ is associated with breast carcinogenesis [30], and a low intake of riboflavin can cause a decrease in the availability of methyl groups, which can result in modifications to the DNA methylation status [31]. Vitamin B_2_ can be found in seafood, meat, algae, pulses, eggs, vegetables, and nuts. The relative bioavailability of vitamin B_2_ to free vitamin B_2_ in the average diet is 64% [32]. Because vitamin B_2_ is a water-soluble vitamin, when saturation is exceeded, urinary excretion increases sharply. Thus, the dietary reference intake for Japanese people argued that there is no evidence of harmful effects of excess intake of vitamin B_2_ on health [6]. The RDA for vitamin B_2_ is 1.2 mg for women aged 18 to 74, 1.0 mg for women aged 75 and above, 1.6 mg for men aged 18 to 49, 1.5 mg for men aged 50 to 74, and 1.3 mg for men aged 75 and above [6]. It is estimated that the RDA satisfies the daily vitamin requirement for almost all individuals in the relevant sex and age groups. According to these criteria, the number of participants in the present study with insufficiency of vitamin B_2_ was 92 (42.2%).

Folic acid is closely related to nucleic acid synthesis and cell proliferation. Hyponutrition of folic acid rises plasma homocysteine, triggering arteriosclerosis [33]. It is difficult to estimate the relative bioavailability of food folic acid; however, a previous study suggested that the bioavailability of folates from food is approximately 80% of that of folic acid [34]. Aging has no effect on the absorption rate of food folic acid in the digestive tract [35]. Thus, the RDA of folic acid is 200 μg for all ages of both sexes [6]. According to these criteria, the number of participants in the present study with folic acid hyponutrient was 48 (22.0%). The tolerable upper intake level of folic acid was not estimated because there are no reports on health damage due to excess intake of folic acid [6]. Previous studies demonstrated that the intake of food folic acid was inversely correlated with the incidence rate of stroke and the mortality rate of cardiovascular disease [36,37]; moreover, folic acid has a preventive effect on stroke and coronary heart disease [38]. Another study reported that the intake of food folic acid was correlated with %BOP, an indicator of current inflammation of a periodontal lesion in non-smoking persons [39]. However, the correlation coefficient was −0.147, which is very low, even though it was statistically significant. In this study, we found no correlation between dietary intake of folic acid and %BOP regardless of smoking habit (Appendix A). The relationship between the intake of food folic acid and periodontal status should be examined in the future.

Vitamin K activates blood coagulation factors, such as prothrombin in the liver [40], osteocalcin to enhance osteogenesis in bone [41], and matrix gla protein (MGP), a small Gla vitamin K-dependent protein, to inhibit vascular calcification [42]. Because enteric bacteria synthesize vitamin K2, it is rare to lack vitamin K, except when receiving intense antimicrobial therapy. In older adults, the absorption rate of vitamin K is diminished due to a decrease in the secretion of bile and pancreatic fluid or a decrease in daily fat intake. The periodontal pathogen *Porphyromonas gingivalis* requires vitamin K for its growth [43]. A previous in vitro study suggested that *Bifidobacterium adolescentis*, a saliva bifidobacteria, may inhibit the growth of *P. gingivalis* by competing to consume the vitamin K in the environment [44]. Systemic administration of vitamin K2 did not affect absorption of alveolar bone or secretion of inflammatory cytokines in a rat model of periodontitis [45]. Thus, the effect of food-driven vitamin K on oral health is still unclear.

The reason why individuals with good oral health behaviors tend to consume higher amounts of vitamins is not yet fully understood. Considerable evidence suggests that individuals with a higher education and income tend to consume greater quantities of green and yellow vegetables [46,47,48,49]. Green and yellow vegetables, such as spinach and broccoli, are rich sources of vitamin B_2_, folic acid, and vitamin K. In the present study, respondents with good oral health behavior consumed more green and yellow vegetables compared to those with poor oral health behavior (86.8 [0–496] vs. 65.6 [0–422] g/day, *p* = 0.009; Appendix A).

Milk and dairy products are good sources of vitamin B_2_. Although not statistically significant, respondents with good oral health behavior tended to consume more milk and dairy products compared to those with poor oral health behavior (140.0 [0–626] vs. 60.0 [0–628] g/day, *p* = 0.069; Appendix A). Additionally, individuals with higher socioeconomic levels tended to consume more milk and dairy products than those with lower socioeconomic levels [49,50]. We observed that the proportion of respondents living in areas with higher levels of education and income was higher in the yes group (i.e., individuals with good oral health behaviors) compared to the proportion in the no group (Table 1). One previous study demonstrated that oral health behavior is related to income level [51], supporting our findings. Based on the above evidence, it can be deduced that individuals with good oral health behaviors may tend to consume higher amounts of vitamins due to their socioeconomic status. However, multiple regression analysis showed an independent correlation between oral health behavior and vitamins B, K, and folic acid intake, even after adjusting for education and income levels. In contrast, vitamins A, E, and niacin were dependent on education and income levels. Hence, dietary behaviors related to food intake containing vitamins B, K, and folic acid, but not other vitamins, may directly correlate with other health-related behaviors, including oral hygiene. 

Dietary counseling or nutrition guidance are essential in dental practice because sugar consumption is linked to the onset of dental caries, and a high fat and high calory diet, associated with the onset of diabetes, increases the risk of exacerbation of periodontitis. The role of dental professionals in public health is not only to prevent and treat oral diseases, but also to act as gatekeepers for the systemic health of community-dwelling individuals [52]. Our finding that oral health behavior correlated with the daily intake of vitamins must be useful for health guidance in multidisciplinary healthcare. A recent meta-analysis revealed that good oral hygiene status, frequent tooth brushing, and frequent interdental cleaning were associated with a lower risk of metabolic syndrome [53]. Oral frailty may precede systemic frailty, and older adults with frailty often exhibit poor oral hygiene status and inadequate oral health behavior [54]. Vitamin insufficiencies are associated with the risk of developing metabolic syndrome [55,56], cardiovascular disease [36,37], type 2 diabetes mellitus [57], and frailty [58]. Therefore, health professionals should consider oral health behavior in collaboration with dental professionals.

The study has some limitations. First, in terms of sociodemographic variables, data regarding the education and income levels corresponded to the settled area; they were not linked to individuals. To establish the robust effect of those characteristics, individual-level data linkage would be preferable. Second, the number of participants was insufficient. Although the data were from a national survey, the number of participants in the national survey was only 6278. In addition, due to the COVID-19 spread, future surveys are likely to continue to have lower numbers of participants. Thus, increasing the number of participants in future surveys is a challenging issue that is difficult to resolve. Third, because this was a cross-sectional study, a causal relationship between the dietary intake of vitamins and oral health behavior was not elucidated. Fourth, there was a difference in the number of teeth between the groups. A recent meta-analysis demonstrated that there is a positive correlation between the number of removed teeth and the risk of developing coronary heart disease [59] or all-cause mortality [60]. The association between the number of teeth that remain, and overall health is crucial and warrants further investigation in future studies. Fifth, it is important to emphasize that simply having a habit of cleaning one’s mouth does not ensure efficient hygiene, nor does it guarantee that the oral microbiota is in a state of balance and that oral health is at an ideal level. The current study did not assess the difference in microbiota between the yes and no groups. Further research is needed to investigate the association between microbiota and vitamin intake.

## 5. Conclusions

Oral health behaviors (e.g., interdental cleaning, tongue brushing) were correlated with the daily intake of vitamins B_2_, K, and folic acid. Our findings can help to reduce the risk of non-communicable diseases or frailty in multidisciplinary healthcare.

## Figures and Tables

**Table 1 ijerph-20-05243-t001:** Characteristics of participants divided based on oral health behavior with interdental cleaning or tongue brushing.

Variables	Oral Health Behavior with Interdental Cleaning or Tongue Brushing		*p*-Value
	Yes (*n* = 133)	No (*n* = 85)	
Age (*m*)	65 (22–90)	64 (22–93)	0.849 ^†^
Sex			
Male (*n*)	61 (45.9)	46 (54.1)	0.235 ^‡^
Female (*n*)	72 (54.1)	39 (45.9)	
Smoking			
Yes (*n*)	22 (16.5)	22 (25.9)	0.094 ^‡^
No (*n*)	111 (83.5)	63 (74.1)	
Education			
≤25% (*n*)	34 (25.6) <–2.9>	38 (44.7) <2.9>	0.007 ^‡^
≤35% (*n*)	76 (57.1) <1.5>	40 (47.1) <–1.5>	
>35% (*n*)	23 (17.3) <1.9>	7 (8.2) <–1.9>	
Income			
1st grade (*n*)	25 (18.8) <–3.1>	32 (37.6) <3.1>	0.004 ^‡^
2nd grade (*n*)	24 (18.0) <–1.0>	20 (23.5) <1.0>	
3rd grade (*n*)	54 (40.6) <2.2>	22 (25.9) <–2.2>	
4th grade (*n*)	30 (22.6) <1.8>	11 (12.9) <–1.8>	
Oral health			
number of teeth (*m*)	13 (0–30)	10 (0–29)	0.006 ^†^
FTU (*m*)	12 (0–12)	12 (1–12)	0.603 ^†^
PPD <4 mm (*m*)	1 (0–6)	0 (0–6)	0.195 ^†^
BOP (*m*)	0 (0–1)	0 (0–1)	0.816 ^†^
Physical examination			
BMI (*m*, kg/m^2^)	22.7 (15.6–35.7)	22.5 (16.7–34.3)	0.983 ^†^
AC (*m*, cm)	82.1 (62.0–114.5)	84.5 (63.0–111.0)	0.652 ^†^
SAP (*m*, mmHg)	130 (88–176)	130 (90–212)	0.647 ^†^
DBP (*m*, mmHg)	78 (50–108)	78 (40–110)	0.493 ^†^
Blood test			
HbA1c (*m*, %)	5.4 (4.6–7.9)	5.5 (4.6–7.4)	0.966 ^†^
Total chol (*m*, mg/dL)	199.5 (131–303)	205.5 (136–327)	0.744 ^†^
HDL chol *m*, (mg/dL)	65.0 (29–112)	60.0 (33–99)	0.028 ^†^
LDL chol (*m*, mg/dL)	118 (60–225)	119.5 (61–246)	0.657 ^†^
Dietary intake			
total protein (*m*, g)	72.3 (23.6–135.7)	69.1(21.7–119.6)	0.049 ^†^
total fat (*m*, g)	62.2 (20.3–141.3)	56.2 (15.9–121.4)	0.034 ^†^
Carbohydrate (*m*, g)	258.3 (131.5–536.4)	258.2 (127.4–528.2)	0.691 ^†^
Sodium (*m*, g)	3.6 (1.1–9.7)	3.8 (1.4–8.0)	0.378 ^†^
Calcium (*m*, mg)	580.7 (113.0–1282.1)	491.6 (116.7–1922.8)	0.016 ^†^
Magnesium (*m*, mg)	251.6 (80.8–752.9)	230.0 (80.3–526.6)	0.078 ^†^
Vitamin A (*m*, µgRAE)	470.7 (43.7–7713.5)	355.4 (41.0–2916.2)	0.001 ^†^
Vitamin B_1_ (*m*, mg)	0.8 (0.3–2.4)	0.8 (0.2–2.0)	0.068 ^†^
Vitamin B_2_ (*m*, mg)	1.2 (0.3–2.9)	1.1 (0.2–2.2)	0.028 ^†^
Vitamin B_6_ (*m*, mg)	1.2 (0.3–3.0)	1.0 (0.3–2.8)	0.017 ^†^
Vitamin B_12_ (*m*, mg)	4.4 (0.4–49.4)	3.2 (0.6–29.3)	0.277 ^†^
Folic acid (*m*, µg)	294.5 (87.1–1262.6)	252.5 (102.4–734.3)	0.003 ^†^
Niacin (*m*, mgNE)	14.3 (3.0–45.5)	12.5 (3.7–41.9)	0.008 ^†^
Vitamin C (*m*, mg)	98.0 (4.5–422.2)	74.3 (8.2–319.9)	0.076 ^†^
Vitamin D (*m*, µg)	4.5 (0.1–38.9)	3.6 (0.1–42.4)	0.216 ^†^
Vitamin E (*m*, mg)	7.1 (2.0–17.8)	6.5 (1.5–15.3)	0.016 ^†^
Vitamin K (*m*, µg)	197.8 (31.8–926.5)	145.4 (16.4–743.8)	0.005 ^†^

^†^: Mann–Whitney U test, ^‡^: chi–square test, number in < > indicates results of residual analysis. PPD: periodontal pocket depth, BMI: body mass index, BOP: bleeding on probing, FTU: functional tooth unit, AC: abdominal circumference, SAP: systolic arterial pressure, DBP: diastolic blood pressure, chol: cholesterol, HbA1c: hemoglobin A1c. HDL: high density lipoprotein, LDL: low density lipoprotein, RAE: retinol activity equivalent, and NE: niacin equivalent.

**Table 2 ijerph-20-05243-t002:** Multiple regression analysis for correlation between dietary intake nutrients and oral health behavior with interdental cleaning or tongue brushing.

Nutrients	Crude Model			Adjusted Model		
	B ± SE	β	*p*-Value	B ± SE	β	*p*-Value
Total protein	(6.4 ± 3.0) × 10^5^	0.15	0.036	(4.5 ± 3.2) × 10^5^	0.10	0.160
Total fat	(6.9 ± 3.3) × 10^5^	0.14	0.037	(4.4 ± 3.3) × 10^5^	0.09	0.186
Calcium	(7.0 ± 3.8) × 10^6^	0.13	0.072	(3.7 ± 4.1) × 10^6^	0.07	0.364
Vitamin A	(2.9 ± 1.3) × 10^7^	0.16	0.024	(2.5 ± 1.4) × 10^7^	0.13	0.074
Vitamin B_2_	(1.7 ± 0.7) × 10^4^	0.17	0.015	(1.6 ± 0.7) × 10^4^	0.16	0.029
Vitamin B_6_	(1.7 ± 0.7) × 10^4^	0.16	0.018	(1.4 ± 0.7) × 10^4^	0.14	0.057
Niacin	(2.0 ± 0.9) × 10^5^	0.15	0.036	(1.3 ± 1.0) × 10^5^	0.10	0.175
Folic acid	(7.7 ± 2.4) × 10^6^	0.22	0.001	(7.0 ± 2.5) × 10^6^	0.20	0.006
Vitamin E	(1.3 ± 4.6) × 10^4^	0.19	0.007	(9.2 ± 4.8) × 10^4^	0.14	0.058
Vitamin K	(7.0 ± 2.6) × 10^6^	0.19	0.007	(5.7 ± 2.8) × 10^6^	0.15	0.043

B: regression coefficient, SE: standard error, β: standardized partial regression coefficient. Adjusted model: adjusted for age, sex, education level of settled area, number of teeth, and abdominal circumference.

## Data Availability

Restrictions apply to the availability of data. Data were obtained from the Ministry of Health, Labor, and Welfare of Japan.

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
