# Peer review of "Correlation between Dietary Intake of Vitamins and Oral Health Behaviors: A Cross-Sectional Study"

_ijerph, 2023, doi:10.3390/ijerph20075243_

Round 1

Reviewer 1 Report (Previous Reviewer 3)

Reviewer comments and suggestions

The authors of this study investigated the correlation between oral health behavior and dietary vitamins intake. This study included respondents aged ≥20 years from the National Health and Nutrition Examination Survey and a national survey of dental diseases in 2016 in Hyogo Prefecture, Japan. The study comprised 218 participants (male: 107, female: 111), aged 64.5 years (range 22–93). The study result included that the daily intakes of protein, fat, calcium, vitamins A, B2, B6, E, and K, folic acid, and niacin were significantly higher in the “yes” group than in the “no” group. Finally, the authors suggested that Oral health behavior with interdental cleaning or tongue brushing correlated with the daily intake of vitamins B2, K, and folic acid.

Overall, the manuscript was well written. However, a few concerns/comments needed to be explained/modified. 

  1. Line 56-57 Please explore the sentence healthy lifestyle cluster.
  2. Line 80-81 It does not stand, do you have any printed version that Governmental approval does not require patient's consent and ethical approval
  3. Line 98 Calculate the power of the study so that we could assume that this number is okay for the analysis
  4. A simple correlation table could be also analyzed with this data, please consult a statistician.
  5. Table 2 First letter should be capitalized.
  6. Line 176-177 is this important to discuss the carcinoma patients with this data
  7. Line 234-235 is there was any relationship with the income trend.
  8. Please check the journal style of references 21,24,34,52

Author Response

Reviewer 2 Report (Previous Reviewer 4)

Overall comment

I have no issue with the results and statistical analysis, but the major shortcoming lies in the language and writing style. The language is difficult to understand and I strongly suggest the paper to undergo for language editing services.

Introduction

The overall introduction needs a major revision and rephrasing. Most sentences do not make any sense.

Line 56-57 “healthy lifestyle cluster (smoking, alcohol consumption, nutrition, exercise, and weight control)”, why is smoking and alcohol consumption under healthy lifestyle cluster?

Result:

In my opinion, there is no need to mention the entire daily nutrient intake figures in sentence form (as we already can see it the table). The authors only need to highlight the figures that show great difference or significance.

Discussion

Again, language editing is required.

Line 231-256 have a very different writing style from the rest of the manuscript, which is exactly what the usual discussion should have.

Author Response

Reviewer 3 Report (New Reviewer)

The article arising from a cross-sectional study has important limitations regarding the implications to oral health, since its main objective is oral health behavior, not having oral clinical variables as support tools. However, their findings can contribute to the conduction of new studies, since the subject is still unclear of results.

One of the findings was the difference between the number of teeth between the evaluated groups. And among the results present in the literature, it may be a possible hypothesis that correlates oral health, and systemic diseases, mainly cardiovascular diseases. If there was no mistake, I did not find information about this possible relationship in the discussion or limitations of the study. I consider it essential to place such a line of evidence.

Still in the discussion, it is important to clarify that the habit of cleaning cannot guarantee that hygiene is carried out efficiently, even that the oral microbiota present is in a condition of biosis, and that oral health is established as ideal. Since the study cannot clarify that the group considered as "yes" has ideal health conditions in terms of its microbiota, it is essential to include this limitation in the article.

Round 2

Reviewer 3 Report (New Reviewer)

The authors made modifications and responded to all suggestions made. Although it is a clinical study with important limitations, the result can guide future studies, and the findings on dietary intake of vitamins and oral health behavior can answer the nutritional profile of a group of patients at high risk of systemic problems.

This manuscript is a resubmission of an earlier submission. The following is a list of the peer review reports and author responses from that submission.

Round 1

Reviewer 1 Report

The biological plausibility between oral health behavior and vitamin intake lacks justification. The objective of the research is not well defined and lacks relevance.

Author Response

Reply: Thank you for this comment. As you indicated, this study was not intended to verify the biological plausibility of a correlation between insufficient vitamin intake and oral health behavior. Instead, this study aimed to investigate whether oral health behavior correlates with the daily intake of vitamins from a behavioral health perspective. Our results show that individuals who choose diets with sufficient vitamins also engage in oral health-related behaviors, strengthening the knowledge of health behavior. To clarify the study’s aim, we have included an additional sentence in the Introduction section as follows:

Regarding the relationship between a healthy lifestyle and diet choices, a previous study demonstrated that the healthy lifestyle cluster (smoking, alcohol consumption, nutrition, exercise, and weight control) was associated with fruit and vegetable consumption and was closely linked to vitamin intake.

Reviewer 2 Report

surely the number of patient must be improved in the future

Author Response

Reply: Thank you for this comment. The number of participants in future studies is expected to decrease due to psychological factors related to residents' concerns about personal information and the spread of COVID-19. Hence, increasing the number of participants in future research might be challenging.

Reviewer 3 Report

Reviewer comments and suggestions

This study aimed to investigate the relationship between oral health behavior and dietary vitamins intake. This study included respondents aged ≥20 years (218 participants) from the National Health and Nutrition Examination Survey, Japan. Participants were divided into two groups based on oral health behavior: the “yes” group, incorporating interdental cleaning or tongue brushing other than tooth brushing, and the “no” group without such behavior. The study suggested that oral health behavior was correlated with the intake of vitamin B2 (p = 0.029), folic acid (p = 0.006), and vitamin K (p = 0.043) after adjusting for possible confounders.

Below are the comments for this paper to be incorporated in the revised version of the manuscript. 

  1. Line 28-29 Please mention a few references.
  2. Lines 45-46 include more points related to (please mention more references).
  3. Line 69-70 please mention the approval number.
  4. How do the authors measure this (BOP)
  5. please include the power analysis in the result section
  6. Could the authors do a few more analysis such as ROC curve or other analysis to enrich your study data
  7. Line 149-150 the authors could include more references here.
  8. Line 202-205 is there was any mechanism for this relationship.

Author Response

Reviewer #3

Reviewer comments and suggestions

This study aimed to investigate the relationship between oral health behavior and dietary vitamins intake. This study included respondents aged ≥20 years (218 participants) from the National Health and Nutrition Examination Survey, Japan. Participants were divided into two groups based on oral health behavior: the “yes” group, incorporating interdental cleaning or tongue brushing other than tooth brushing, and the “no” group without such behavior. The study suggested that oral health behavior was correlated with the intake of vitamin B2 (p = 0.029), folic acid (p = 0.006), and vitamin K (p = 0.043) after adjusting for possible confounders.

Below are the comments for this paper to be incorporated in the revised version of the manuscript.

Line 28-29 Please mention a few references.

Reply: We have included three additional references for this sentence.

Lines 45-46 include more points related to (please mention more references).

Reply: We have included three additional references for this sentence.

Line 69-70 please mention the approval number.

Reply:

How do the authors measure this (BOP)

Reply: Based on this comment, we have added the following sentences in the revised manuscript: Probing was performed using the WHO probe by sliding the probe’s tip along the surface of the teeth with light pressure (20g). BOP was considered present if observed within 10–30 seconds after probing.

please include the power analysis in the result section

Reply: Based on this suggestion, the following sentences have been added in the revised manuscript: For the multiple regression analysis, when the effect size (f 2) and β/α ratio were 0.15 and 4, respectively, power was calculated as 0.989.

Could the authors do a few more analysis such as ROC curve or other analysis to enrich your study data.

Reply: Thank you for this suggestion. A ROC analysis is useful for evaluating the accuracy of model predictions. We attempted a ROC analysis to predict vitamin intake, but oral health behavior did not match the ROC model. We have added a supplemental table for differences in food consumption based on oral health behavior.

Line 149-150 the authors could include more references here.

Reply: We have included additional references for each biological benefit of vitamin B2 in the sentence as follows; antioxidant (Olfat et al. 2022), anti-aging (Zou et al. 2017), anti-inflammatory (Wald et al. 2022), antinociceptive (Bertollo et al. 2006), and anticancer properties (França et al. 2001).

Line 202-205 is there was any mechanism for this relationship.

Reply: Thank you for this comment. We have revised the text as follows; Insufficiencies of vitamins are associated with risks of metabolic syndrome, cardiovascular disease, type 2 diabetes mellitus, and frailty. Therefore, health professionals should consider oral health behavior in collaboration with dental professionals.

Reviewer 4 Report

Overall Comments:

I felt that this research struggled to overcome the ‘so what’ factor – why is this research important and what is the benefit of this work being published? I would encourage you to consider this as you revise the paper.

Abstract

The abstract is concise and well written. It summarizes the study well.

Materials and Methods

-The authors did not clearly explain how the study ended up with 218 samples. Is a sample size calculation done? Or is this the total number of subject available in the national survey?

-The authors mentioned that the respondents were selected on a household basis. Was it conveniently sampled or a probability sampling technique was employed?

Results

I would like to ask the authors for clarifying the rationale of including physical examination and blood test in Table 1, since the study only aims to investigate the correlation between oral health behaviors and daily intake of vitamin.

Discussion

The major shortcoming of the discussion is that it fails to discuss on the results obtained in this study. To me, it seems more like an overview of vitamins B2, K, and folic acid. I would suggest the authors to discuss why some vitamins intakes are correlated with good oral health behavior while some vitamins are not especially in the Japanese context.

Author Response

Reviewer #4

Overall Comments:

I felt that this research struggled to overcome the ‘so what’ factor – why is this research important and what is the benefit of this work being published? I would encourage you to consider this as you revise the paper.

Abstract

The abstract is concise and well written. It summarizes the study well.

Materials and Methods

-The authors did not clearly explain how the study ended up with 218 samples. Is a sample size calculation done? Or is this the total number of subject available in the national survey?

Reply: As you have observed, the total number of participants available in the national survey was included in this study. We have included the following sentence in the materials and methods: This study included 218 participants, which was the total number of participants available in the national survey for Hyogo prefecture.

-The authors mentioned that the respondents were selected on a household basis. Was it conveniently sampled or a probability sampling technique was employed?

Reply: Data collection per-household basis was legally mandated because the data were obtained from a national survey.

Results

I would like to ask the authors for clarifying the rationale of including physical examination and blood test in Table 1, since the study only aims to investigate the correlation between oral health behaviors and daily intake of vitamin.

Reply: Thank you for this suggestion. It is necessary to consider the possibility of confounding variables, such as physical examinations or blood tests, which have been included in the analysis.

Discussion

The major shortcoming of the discussion is that it fails to discuss on the results obtained in this study. To me, it seems more like an overview of vitamins B2, K, and folic acid. I would suggest the authors to discuss why some vitamins intakes are correlated with good oral health behavior while some vitamins are not especially in the Japanese context.

Reply: Thank you for this suggestion. Based on it, we have included additional discussions on the correlation between oral health behavior and vitamin intake. The following text has been included in the discussion:

The exact reason individuals with good oral health behavior tend to consume higher amounts of vitamins is not yet fully understood. Considerable evidence suggests that individuals with higher education and income tend to consume greater quantities of green and yellow vegetables. Green and yellow vegetables, such as spinach and broccoli, are rich sources of vitamin B2, folic acid, and vitamin K. In the present study, respondents with good oral health behavior consumed more green and yellow vegetables compared to those with poor oral health behavior (86.8 [0–496] vs. 65.6 [0–422] g/day, p = 0.009; supplemental table 2). 

Milk and dairy products are good sources of vitamin B2. Although not statistically significant, respondents with good oral health behavior tended to consume more milk and dairy products compared to those with poor oral health behavior (140.0 [0–626] vs. 60.0 [0–628] g/day, p = 0.069; supplemental table 2). Additionally, individuals with higher socioeconomic levels tended to consume more milk and dairy products than those with lower socioeconomic levels. This study observed that the proportion of respondents living in areas with higher levels of education and income was higher in the “yes” group (i.e., individuals with good oral health behavior) compared to the proportion in the “no” group (Table 1). One previous study demonstrated that oral health behavior is related to income level, supporting our findings. Based on the above evidence, it can be deduced that individuals with good oral health behaviors tend to consume higher amounts of vitamins might be due to their socioeconomic status. However, the multiple regression analysis showed an independent correlation between oral health behavior and vitamins B, K, and folic acid intake, even after adjusting for education and income levels. In contrast, vitamins A, E, and niacin were dependent on education and income levels. Hence, dietary behaviors related to food intake containing vitamins B, K, and folic acid, but not other vitamins, may directly correlate with other health-related behaviors, including oral hygiene. (Lines 215–239)

  1. Ames, B.N. Low micronutrient intake may accelerate the degenerative diseases of aging through allocation of scarce micronutrients by triage. Proc Natl Acad Sci U S A 2006, 103, 17589-17594.
  2. McGuire, S. U.S. Department of Agriculture and U.S. Department of Health and Human Services, Dietary Guidelines for Americans, 2010. 7th Edition, Washington, DC: U.S. Government Printing Office, January 2011. Adv Nutr 2011, 2, 293-294.
  3. Troesch, B.; Hoeft, B.; McBurney, M.; Eggersdorfer, M.; Weber, P. Dietary surveys indicate vitamin intakes below recommendations are common in representative Western countries. Br J Nutr 2012, 108, 692-698.
  4. Sheiham, A.; James, W.P.T. Diet and Dental Caries:The Pivotal Role of Free Sugars Reemphasized. Journal of Dental Research 2015, 94, 1341-1347.
  5. Moynihan, P.; Petersen, P.E. Diet, nutrition and the prevention of dental diseases. Public Health Nutrition 2004, 7, 201-226.
  6. Tanaka, K.; Miyake, Y.; Sasaki, S.; Ohya, Y.; Matsunaga, I.; Yoshida, T.; Hirota, Y.; Oda, H. Relationship between intake of vegetables, fruit, and grains and the prevalence of tooth loss in Japanese women. J Nutr Sci Vitaminol (Tokyo) 2007, 53, 522-528.
  7. Olfat, N.; Ashoori, M.; Saedisomeolia, A. Riboflavin is an antioxidant: a review update. Br J Nutr 2022, 128, 1887-1895.
  8. Zou, Y.X.; Ruan, M.H.; Luan, J.; Feng, X.; Chen, S.; Chu, Z.Y. Anti-Aging Effect of Riboflavin Via Endogenous Antioxidant in Fruit fly Drosophila Melanogaster. J Nutr Health Aging 2017, 21, 314-319.
  9. Wald, E.L.; Badke, C.M.; Hintz, L.K.; Spewak, M.; Sanchez-Pinto, L.N. Vitamin therapy in sepsis. Pediatr Res 2022, 91, 328-336.
  10. Bertollo, C.M.; Oliveira, A.C.; Rocha, L.T.; Costa, K.A.; Nascimento, E.B., Jr.; Coelho, M.M. Characterization of the antinociceptive and anti-inflammatory activities of riboflavin in different experimental models. Eur J Pharmacol 2006, 547, 184-191.
  11. França, D.S.; Souza, A.L.S.; Almeida, K.R.; Dolabella, S.l.S.; Martinelli, C.; Coelho, M.M. B vitamins induce an antinociceptive effect in the acetic acid and formaldehyde models of nociception in mice. European Journal of Pharmacology 2001, 421, 157-164.

Bertollo CM, Oliveira AC, Rocha LT, Costa KA, Nascimento EB, Jr., Coelho MM. 2006. Characterization of the antinociceptive and anti-inflammatory activities of riboflavin in different experimental models. Eur J Pharmacol. 547(1-3):184-191.

França DS, Souza ALS, Almeida KR, Dolabella SlS, Martinelli C, Coelho MM. 2001. B vitamins induce an antinociceptive effect in the acetic acid and formaldehyde models of nociception in mice. European Journal of Pharmacology. 421(3):157-164.

Olfat N, Ashoori M, Saedisomeolia A. 2022. Riboflavin is an antioxidant: A review update. Br J Nutr. 128(10):1887-1895.

Wald EL, Badke CM, Hintz LK, Spewak M, Sanchez-Pinto LN. 2022. Vitamin therapy in sepsis. Pediatr Res. 91(2):328-336.

Zou YX, Ruan MH, Luan J, Feng X, Chen S, Chu ZY. 2017. Anti-aging effect of riboflavin via endogenous antioxidant in fruit fly drosophila melanogaster. J Nutr Health Aging. 21(3):314-319.
